# Granzyme B PET Imaging Enables Early Assessment of Immunotherapy Response in a Humanized Melanoma Mouse Model

**DOI:** 10.3390/ph18091309

**Published:** 2025-08-31

**Authors:** Priska Summer, Naomi Gallon, Niklas Bulmer, Umar Mahmood, Pedram Heidari

**Affiliations:** Division of Nuclear Medicine and Molecular Imaging, Department of Radiology, Massachusetts General Hospital, Boston, MA 02114, USA

**Keywords:** granzyme B PET imaging, molecular imaging, checkpoint inhibitor therapy, humanized mouse model

## Abstract

**Background/Objectives**: This study evaluated a novel PET tracer, ^68^Ga-NOTA-CYT-200, which targets human granzyme B (GZB) as a biomarker for cytotoxic T-cell activation in a clinically relevant model of melanoma-bearing mice with a humanized immune system treated with immune checkpoint inhibitor (ICI) therapy. **Methods**: The binding affinity of the tracer was determined using an enzymatic colorimetric assay. Tumor-bearing humanized NSG mice underwent PET imaging before and during ICI monotherapy or combination therapy to assess ^68^Ga-NOTA-CYT-200 uptake within tumors and other organs. The tumor growth was carefully monitored. The treatment response was evaluated based on the percentage change in tumor size at days 5 and 15 after the treatment started. A tracer biodistribution study and immunohistochemical staining of the tumors and organs were also performed. Results: The inhibition constant (Ki) of ^68^Ga-NOTA-CYT-200 was estimated at 4.2 nM. PET imaging showed a significantly higher ^68^Ga-NOTA-CYT-200 uptake in mice receiving the combination therapy compared to those receiving monotherapy or a vehicle (*p* < 0.0001 or *p* = 0.0005, respectively), which correlated with the greatest reduction in tumor size in the combination ICI group. Regardless of treatment, the responders presented with a significantly higher ^68^Ga-NOTA-CYT-200 uptake at days 4 or 7 after the treatment began (*p* = 0.0002 and *p* = 0.0109, respectively). An increased uptake of ^68^Ga-NOTA-CYT-200, especially in the intestines and liver within the combination ICI group, suggested immune-related adverse events (IrAEs). **Conclusions**: Our study demonstrates that ^68^Ga-NOTA-CYT-200 PET imaging can predict the early treatment response in melanoma models treated with ICI and may also help in detecting IrAEs.

## 1. Introduction

Over the past decade, immunotherapies have revolutionized cancer treatment [1,2,3,4]. The proportion of cancer patients eligible for immune checkpoint inhibitor (ICI) therapy has grown significantly, increasing 30-fold over nine years—from 1.54% in 2011 (95% CI: 1.51–1.57%) to 43.63% in 2018 (95% CI: 43.51–43.75%) [5]. However, ICI therapies still face some important limitations. The most significant issue is the response rates of less than 50% for many cancers, such as melanoma, urothelial carcinoma, and non-small cell lung cancer [1]. Conversely, the development of immunotherapy-related adverse events (IrAEs), especially with an ICI combination regimen, is linked to poorer treatment outcomes, as they can lead to severe, hard-to-manage health conditions that may result in treatment discontinuation or the patient’s death [2,3]. Overall, there is an urgent need for a reliable tool to quickly and accurately assess patient response to ICI therapy. To date, significant efforts have been made to develop a non-invasive method for evaluating the therapeutic effectiveness and the early treatment response in immunotherapy. Several potential biomarkers for response prediction, including PD-1, PD-L1, CD8, and CD3, are being studied through various methods, such as biopsies, genetic screening, and molecular imaging [4,5,6,7]. The current biomarkers primarily rely on biopsy samples, but intra-tumoral immune cell heterogeneity may limit their predictive value [1]. Additionally, tumoral immune cell infiltrates can cause tumors to appear to grow on imaging modalities like CT and MRI—a phenomenon known as pseudo-progression—reducing the utility of early measurement [2]. Molecular imaging offers a promising alternative by enabling a quantitative assessment of biomarkers across the entire tumor burden, providing a logical approach to capturing tumor biology. For instance, granzyme B (GZB), a key effector molecule involved in tumor cell death, serves as a valid biomarker for T-cell-mediated cytotoxicity and the anti-tumor response [3,4]. Specifically, GZB PET imaging has emerged as a promising method for detecting the immunotherapy response [5]. Unlike biopsies, which are invasive and limited by sampling bias and procedural risks, molecular imaging allows for non-invasive, whole-body, and serial assessments. PET imaging provides a three-dimensional, functional overview of molecular processes, capturing early biochemical and cellular changes before anatomical alterations become evident [6]. It is particularly advantageous in immunotherapy, where immune activation and toxicity can occur at multiple sites. Its sensitivity enables real-time monitoring throughout the treatment, addressing the spatial and temporal heterogeneity of the immune responses [7]. PET has a lower spatial resolution than histology and requires highly specific tracers to minimize non-target binding. Still, its ability to visualize disease dynamics in vivo drives the ongoing development of more selective radiotracers for precise immune monitoring.

This study aimed to thoroughly evaluate the usefulness of a new human GZB PET imaging tracer, ^68^Ga-NOTA-CYT-200, in assessing the responses to ICI therapy by comparing the in vivo tumor and organ radiotracer uptake with ex vivo data such as a biodistribution and immunohistochemical analysis. ^68^Ga-NOTA-CYT-200 is a novel PET probe designed to target the active form of human GZB; it consists of a GZB-targeting peptide linked to a NOTA chelator, allowing efficient radiolabeling with gallium-68 for PET imaging. NOTA-CYT-200 has a nanomolar inhibition constant (Ki 4.2 nM) for GZB, indicating strong, selective binding to active GZB with minimal off-target effects. Its design facilitates quick clearance from non-target tissues, boosting the tumor-to-background ratios and imaging accuracy. Compared to earlier GZB-targeting agents, NOTA-CYT-200 was selected for its improved specificity, better pharmacokinetics, and suitability for humanized models, making it an effective tool for detecting functional cytotoxic T-cell activity during the effector phase. The early detection of active immune responses—potentially before observable tumor changes—is a key advantage of this tracer, which is the focus of this study.

## 2. Results

### 2.1. ^68^Ga-NOTA-CYT-200 Has a High Target Specificity for GZB

The human GZB enzymatic inhibition assay showed that NOTA-CYT-200 inhibited active GZB activity in a concentration-dependent manner. As the concentration of NOTA-CYT-200 increased, the conversion of the substrate to a fluorescent molecule by GZB decreased, resulting in a lower fluorescence intensity measured in relative fluorescence units (RFUs). Based on this, the inhibition constant (Ki) of NOTA-CYT-200 for active GZB was estimated at 4.2 nM (CI: 2.95–5.98) (Figure 1B). PET imaging with ^68^Ga-NOTA-CYT-200 in mice implanted with Matrigel containing active GZB protein (n = 8) and Matrigel without the protein (n = 8; Figure 1C) showed a significant difference in radiotracer uptake (6.18 ± 2.92 vs. 2.48 ± 1.01, *p* = 0.0045) (Figure 1D).

### 2.2. ^68^Ga-NOTA-CYT-200 PET Uptake Is Related to the Treatment Regimen

We observed that the ^68^Ga-NOTA-CYT-200 PET uptake was related to the treatment regimen at days 4 and 7 after therapy initiation. At baseline PET imaging, there was no significant difference in the TBR among the treatment groups (controls, n = 6: 1.35 ± 0.2; monotherapy, n = 9: 1.25 ± 0.17; combination, n = 8: 1.47 ± 0.2, *p* > 0.85). Four days after starting treatment, the combination group (n = 8) exhibited a significantly higher mean TBR compared to the controls (n = 6) (3.88 ± 0.47 vs. 1.15 ± 0.13, *p* < 0.0001), as did the monotherapy group (n = 9) (2.84 ± 0.3, *p* = 0.0005). The combination group (n = 8) also had a significantly higher TBR than the monotherapy group (n = 9) (3.88 ± 0.47 vs. 2.84 ± 0.3, *p* = 0.027). At 7 days post-treatment, the combination group (n = 8) still showed a significantly increased radiotracer activity compared to the monotherapy group (n = 9) (3.5 ± 0.48 vs. 2.14 ± 0.11, *p* = 0.0052) and the controls (n = 6) (1.69 ± 0.2, *p* = 0.0003), and a significant difference was no longer observed between the monotherapy group (n = 9) and the controls (n = 6) (2.14 ± 0.11 vs. 1.69 ± 0.2, *p* = 0.59). At the final PET imaging timepoint (day 14), the difference between the combination (n = 8) and monotherapy (n = 9) groups was less marked than at earlier timepoints (3.05 ± 0.43 vs. 1.75 ± 0.15, *p* = 0.01), as was the difference between the combination group (n = 8) and the controls (n = 6) (3.05 ± 0.43 vs. 1.97 ± 0.28, *p* = 0.05). There was no significant difference between the monotherapy group (n = 9) and the controls (n = 6) (1.75 ± 0.15 vs. 1.97 ± 0.28, *p* = 0.88). Overall, the combination group (n = 8) showed the most significant increase in the TBR compared to baseline imaging (1.47 ± 0.2 vs. 3.88 ± 0.47, *p* < 0.0001), followed by the monotherapy group (n = 9) (1.25 ± 0.17 vs. 2.84 ± 0.3, *p* = 0.05), while the controls (n = 6) exhibited similar TBR values 4 days after the treatment compared to baseline (1.35 ± 0.18 vs. 1.15 ± 0.13, *p* > 0.99; Figure 2). The increase in the TBR in the controls (n = 6) over time was due to the PBMC reconstitution model, which shows low-level immune activation as the human immune system engrafts and responds to murine tissues.

### 2.3. The Combination Treatment Led to the Most Significant Tumor Size Reduction

The tumor volumes for each treatment group were measured regularly every two to three days for up to fifteen days after the initial treatment (Figure 3A). As early as 5 days post-treatment, both the monotherapy group (n = 15) and the combination group (n = 15) showed significantly smaller tumor volumes compared to the controls (n = 12) (controls: 238.65 ± 54.28 mm^3^ vs. monotherapy: 94.62 ± 38.81 mm^3^, *p* = 0.0002; controls vs. combination: 144.82 ± 43.10 mm^3^, *p* = 0.0255). There was no significant difference between the monotherapy (n = 15) and combination (n = 15) groups at this point. On day 7, this difference became more evident, with both treatment groups again exhibiting significantly smaller tumor volumes compared to the controls (n = 12) (controls: 261.87 ± 64.69 mm^3^ vs. monotherapy: 137.02 ± 49.42 mm^3^, *p* = 0.0013; controls vs. combination: 103.77 ± 67.67 mm^3^, *p* < 0.0001). No significant difference was seen between the monotherapy (n = 15) and combination (n = 15) groups. By day 15, one mouse in the combination group (n = 14) had to be euthanized, but this group still maintained a statistically significant reduction in tumor volume compared to the controls (n = 12) (controls: 259.85 ± 76.28 mm^3^ vs. combination: 99.26 ± 43.67 mm^3^). Meanwhile, the tumors in the monotherapy group (n = 15) had begun to regrow, resulting in no significant difference compared to the controls (n = 12) at this later time (controls: 259.85 ± 76.28 mm^3^ vs. monotherapy: 219.28 ± 62.73 mm^3^), as shown in Figure 3A–C.

Overall, these findings confirm that the combination group (n = 14) exhibited the most pronounced and durable reduction in tumor burden, while the group receiving monotherapy (n = 15) also induced significant, but less sustained, tumor suppression compared to the controls (n = 12).

### 2.4. ^68^Ga-NOTA-CYT-200 PET Imaging Can Differentiate Treatment Responders from Non-Responders Before Tumor Volume Changes

To identify responders (TRs) and non-responders (TNRs) in groups receiving combination therapy (n = 8) or monotherapy (n = 9), changes in the tumor size (%) at days 5 and 15 after the start of treatment were compared to baseline. Mice with tumors that showed a reduction in size at both timepoints (day 5 and day 15) were classified as TRs. Conversely, mice that exhibited a tumor size reduction at only one—typically the earlier—timepoint or no reduction at all were classified as TNRs. Based on this, three out of nine mice receiving monotherapy were classified as TRs and six as TNRs (TRs, n = 3; TNRs, n = 6). In the group receiving combination therapy (n = 15; subset imaged, n = 8), six mice were TRs and one was a TNR (TRs, n = 6; TNRs, n = 1). One mouse in the combination group showed a 50% reduction in tumor volume five days after the treatment began, but had to be euthanized before day 15 and was therefore not included. The tumor volumes of these subgroups were compared on days 0, 5, 10, and 15 after treatment initiation (Figure 4A). The tumor volumes of the controls (n = 12; 223 ± 87 mm^3^), TNRs (n = 7 across treatment groups; 221 ± 95 mm^3^), and TRs (n = 9 across treatment groups; 164 ± 88 mm^3^) were not significantly different (*p* > 0.05) at baseline. While the TNRs (n = 7) and TRs (n = 9) showed a significant tumor size reduction five days after treatment compared to the controls (n = 12) (controls: 262 ± 64 mm^3^ vs. TNRs: 108 ± 50 mm^3^, *p* = 0.001; vs. TRs: 105 ± 43 mm^3^, *p* = 0.001), the average tumor volume of the TNRs increased on days 10 and 15, becoming similar to that of the controls by day 15 (controls: 301 ± 57 mm^3^ vs. TNRs: 311 ± 121 mm^3^) (Figure 4B). However, the TRs (n = 9) exhibited significantly lower tumor volumes (97 ± 47 mm^3^) 15 days after the treatment started compared to the TNRs (n = 7, *p* < 0.0001) and controls (n = 12, *p* < 0.0001). Compared to the controls (n = 6, 1.1 ± 0.3), the TBR measured during PET imaging four days after treatment began was significantly greater for the TNRs (n = 6, 2.7 ± 0.9, *p* = 0.02) and TRs (n = 3, 4.0 ± 1.2, *p* < 0.0001). We then selected the highest ^68^Ga-NOTA-CYT-200 TBR for each mouse’s tumors on either day 4 or day 7 after the treatment start to see if there was a difference between TRs and TNRs, even before changes in the tumor volume could be observed. The TBRs were not significantly different for the controls (n = 6) and TNRs (n = 6) (1.75 ± 0.39 vs. 2.73 ± 0.88, *p* = 0.175). However, there was a significant difference between the controls (n = 6) and TRs (n = 3) (1.75 ± 0.39 vs. 4.28 ± 1.20, *p* = 0.0002) and between the TNRs (n = 6) and TRs (n = 3) (2.73 ± 0.88 vs. 4.28 ± 1.20, *p* = 0.0109).

### 2.5. Treated Mice Showed Greater GZB Expression in the Liver and Intestines

The measurements of the ^68^Ga-NOTA-CYT-200 PET uptake, evaluated by the intestinal/colon-to-blood ratios (CBR) and liver-to-blood ratios (LBR), were consistently taken from comparable regions of the liver and intestines across all the groups. Initially, the CBR values were similar among the control (n = 6; 2.11 ± 0.56), monotherapy (n = 9; 2.68 ± 0.43), and combination groups (n = 8; 3.5 ± 0.62), with no significant differences observed (*p* > 0.99). The combination group (n = 8) showed a significant increase in the CBR on day 4 compared to baseline (6.81 ± 1.21 vs. 3.5 ± 0.62, *p* = 0.029), which then gradually decreased on days 7 (6.21 ± 1.23) and 14 (4.09 ± 0.97), without significant differences from baseline (*p* = 0.96 and *p* = 0.99, respectively). The monotherapy group (n = 9) showed the highest CBR at day 4 (5.41 ± 0.98), followed by reductions on days 7 (3.75 ± 0.57) and 14 (2.25 ± 0.36), but none of these changes were statistically significant relative to baseline (day 4: *p* = 0.07; day 7: *p* = 0.8; day 14: *p* = 0.99). The control subjects (n = 6) maintained stable CBR values throughout the imaging period, ranging from 1.68 ± 0.47 to 2.58 ± 1.17 (*p* > 0.99).

The baseline LBRs were also comparable across all the groups (monotherapy [n = 9] vs. combination [n = 8], *p* = 0.90; monotherapy vs. controls [n = 6], *p* = 0.99). Following treatment, the combination group (n = 8) exhibited significantly elevated LBRs on days 4 (6.93 ± 1.04) and 7 (7.16 ± 1.03) compared to baseline (2.98 ± 0.34; *p* < 0.0001 for both). By day 14 (3.52 ± 0.27), the LBR declined significantly relative to days 4 and 7 (*p* = 0.0004 and *p* = 0.0001, respectively) and returned to levels similar to the pre-treatment values (*p* = 0.91). The monotherapy group (n = 9) showed a significant LBR increase on day 4 versus baseline (*p* = 0.023), while days 7 and 14 remained comparable to baseline (*p* = 0.93 and *p* = 0.71), with a significant decrease from day 4 to 14 (*p* = 0.027). The controls (n = 6) demonstrated a higher post-treatment LBR on day 4 compared to day 14 (*p* = 0.04), though none of the post-treatment LBR values were significantly different from baseline (*p*-values ranging from 0.43 to 0.71). Figure 5 visually presents the dynamic changes in the ^68^Ga-NOTA-CYT-200 uptake in the intestines and liver, demonstrating peak tracer accumulation early after combination therapy and a reduced uptake over time, which is suggestive of treatment-induced immune activation and potential IrAEs.

### 2.6. Ex Vivo Measurements of GZB Correlate with In Vivo Findings

Four days after starting treatment was identified as the peak point for in vivo tracer uptake, indicating the highest GZB activity and immune cell cytotoxic response within the tumor microenvironment. Consequently, we measured the biodistribution of ^68^Ga-NOTA-CYT-200 one hour post-injection at this time. The data revealed the kidney as the main organ responsible for tracer elimination, with uptake values of 1.62 ± 0.22 (%ID/g) in the combination group (n = 3), 1.47 ± 0.37 in the monotherapy group (n = 2), and 1.32 ± 0.84 in the controls (n = 3). Other organs exhibited a lower tracer uptake, similar to the heart. In line with the in vivo tumor-to-blood ratio (TBR) measurements, tracer accumulation within tumors was the highest in the combination group (0.50 ± 0.17%ID/g, n = 3), compared to the monotherapy group (0.25 ± 0.06, n = 2) and the controls (0.12 ± 0.03, n = 3). However, these differences were not statistically significant (combination vs. controls: *p* = 0.49; monotherapy vs. controls: *p* = 0.92; combination vs. monotherapy: *p* = 0.71). Notably, tracer accumulation in the intestines was significantly greater in the combination group (1.12 ± 0.67, n = 3) than in the monotherapy group (0.17 ± 0.66, n = 2, *p* = 0.01) or the controls (0.17 ± 0.13, n = 3, *p* = 0.02), with no significant difference between the monotherapy group and the controls (*p* > 0.99). Although the liver tracer activity was also highest in the combination group (0.57 ± 0.19, n = 3), this increase was not statistically significant when compared to the monotherapy group or the controls (*p* = 0.46 and *p* = 0.63, respectively).

To further confirm the PET imaging results, immunofluorescent GZB staining was conducted on tumor, intestinal, and liver tissues collected from all groups on day 4 after treatment. In tumors, the GZB expression was highest for the combination group (49.59 ± 11.99, n = 3), significantly exceeding that of the onotherapy group (3.31 ± 1.19, n = 2, *p* = 0.03) and the controls (0.06 ± 0.04, n = 3, *p* = 0.01). The monotherapy group showed a slight, non-significant increase compared to the controls (*p* = 0.96). Similarly, the GZB levels in the intestinal tissues were greatest in the combination group (23.8 ± 0.68, n = 3), followed by the monotherapy group (19.59 ± 2.61, n = 2) and the controls (17.13 ± 3.35, n = 3), though these differences were not statistically significant (combination vs. monotherapy: *p* = 0.54; combination vs. controls: *p* = 0.21; monotherapy vs. controls: *p* = 0.8). The liver GZB staining was increased in the monotherapy (60.42 ± 31.91, n = 2) and combination (49.99 ± 12.97, n = 3) groups. Although these groups had slightly higher staining than the controls (19.27 ± 5.01, n = 3), the differences lacked statistical significance (*p* = 0.27 and *p* = 0.37, respectively; Figure 6).

## 3. Discussion

To our knowledge, this is the first study to utilize a humanized mouse model that incorporates both human melanoma cells (G361) and human PBMCs, creating a live experimental system with a functional human tumor-immune microenvironment, which enhances its translational relevance. Unlike traditional murine models that lack human immune components, this approach more accurately mimics human immunotherapy responses. The G361 melanoma cell line, known for high GZB expression and responsiveness to immune therapies [7], was specifically chosen for this study because of its suitability for the methodology used.

In this study, we used a novel PET imaging probe, ^68^Ga-NOTA-CYT-200, which targets active human GZB as a biomarker for cytotoxic T-cell activity in a humanized mouse model of melanoma undergoing ICI therapy. Notably, as shown in our enzymatic inhibition assays, NOTA-CYT-200 can directly inhibit active GZB with a low nanomolar inhibition constant (Ki 4.2 nM). Compared to previous GZB PET probes [4], NOTA-CYT-200 has enhanced binding due to an optimized molecular structure, resulting in a higher sensitivity and a better signal-to-background ratio for detecting effector immune activity. Our results demonstrate that ^68^Ga-NOTA-CYT-200 binds specifically and with a high affinity to active GZB in both in vitro and in vivo settings, with significantly greater tracer activity in GZB-containing Matrigel implants compared to those without GZB.

Larimer et al. investigated murine GZB PET probes, such as ^68^Ga-NOTA-GZP, in syngeneic mouse models [5]. They found that responders to combination anti-PD-1/anti-CTLA-4 immunotherapy exhibited TBR values of around 1.83 ± 0.18, compared to 1.29 ± 0.12 with monotherapy and 0.96 ± 0.11 for controls. Notably, mice with a TBR above 1.9 were identified as responders even before clear tumor shrinkage was observed, while non-responders and controls showed a consistently lower uptake. Treatment responders had TBRs of 1.90 ± 0.55 versus 0.89 ± 0.19 in non-responders; the controls averaged 0.95 ± 0.20 [4]. Their subsequent work further showed that both the effectiveness of checkpoint inhibitor combinations and the impact of administration timing can be quantified using GZB PET imaging, thus providing additional translational support for our observed imaging phenomena [8]. Recent reports describe similar ranges for the maximum SUV, typically 0.75–1.9 in both preclinical and early clinical models. Our maximum TBR values—4.28 ± 1.20 for responders and 2.73 ± 0.88 for non-responders—are higher than those previously reported for GZB-targeted PET imaging, reflecting the robust GZB-mediated immune activity in this humanized model and the potentially favorable affinity of the ^68^Ga-NOTA-CYT-200 peptide (Ki, 4.2 nM). While such a high affinity enhances the imaging sensitivity and specificity, it also raises questions about possible biological implications. Theoretically, the inhibition of GZB could, at sufficient concentrations, reduce the proteolytic activity of this enzyme, potentially modulating T-cell-mediated cytotoxicity if used at high doses. However, in PET imaging, the tracer is administered at doses far below pharmacologically active levels, making significant inhibition unlikely. The overall exposure during imaging is orders of magnitude below what would be necessary to interfere with the endogenous GZB function, minimizing the risk of immunosuppression or immune response interference. Nonetheless, this concern underscores the importance of careful dose management in future therapeutic applications or repeated imaging. In our study, we did not observe any obvious inhibition of the inflammatory response to immunotherapy as a result of a ^68^Ga-NOTA-CYT-200 injection, despite multiple-timepoint imaging. However, further preclinical and clinical safety studies are essential to exclude off-target effects due to enzymatic inhibition confidently.

PET imaging on tumor-bearing mice was performed on days 4, 7, and 14 following treatment initiation. These specific timepoints were selected based on the established kinetics of cytotoxic lymphocyte activation, GZB activity, and the tumor response in both preclinical and clinical studies of ICI therapy [5,9,10]. Early after the initiation of anti-PD-1 or anti-CTLA-4 therapy, effector T-cell activation and GZB release are reported to begin within 3–5 days, preceding observable tumor shrinkage [5,11,12]. Therefore, imaging at day 4 was chosen to capture this initial phase of immune activation. Day 7 imaging corresponds to the peak phase of immune effector expansion and granzyme B secretion, which numerous murine models have demonstrated occurs within the first 1–2 weeks after s checkpoint blockade [5,10]. At this stage, differences in the PET signal between responders and non-responders are often most pronounced. In this study, the TBR measured four to seven days after therapy initiation was significantly greater in the responders (4.28 ± 1.20) than in the non-responders (2.73 ± 0.88) and controls (1.75 ± 0.39), thereby strongly aligning with other preclinical GZB PET imaging data [9,13].

To evaluate the duration of immune activation caused by the treatment and differentiate between sustained and transient responses, day 14 was selected as a late timepoint. Previous studies have shown that signal declines occur in non-responders after an initial surge of immune activation, often due to T-cell exhaustion or adaptive resistance [7,10]. Together, the timepoints (days 4, 7, and 14) were chosen to thoroughly assess the initiation, peak, and persistence of anti-tumor immune activity and align with prior research using GZB-targeted PET imaging to identify response patterns [4,11]. As anticipated, combining two ICIs was more effective at reducing the tumor size than monotherapy [12], since a stronger T-cell response is expected with combination treatments [13]. The group receiving the combination therapy showed the greatest increase in tumoral ^68^Ga-NOTA-CYT-200 uptake four days after the treatment started, represented by a TBR of 3.88 ± 0.47 versus 2.84 ± 0.3 (monotherapy) and 1.15 ± 0.13 (controls). Our findings show an inverse relationship between the TBR and tumor growth in mice treated with both combination therapy and monotherapy. Notably, the combination therapy group experienced a significant reduction in tumor size compared to the controls by day seven after the treatment (–64.31 ± 7.71% vs. 17.59 ± 13.97% change from baseline, *p* = 0.0049), and this difference remained significant up to day 15. Consistent with previous research [5], a temporary decrease in tumor volume was observed in the monotherapy group within the first five days, followed by regrowth. The controls exhibited steady tumor growth across all timepoints. This pattern is consistent with a brief reactivation of exhausted cytotoxic T-cells following the PD-1 checkpoint blockade. However, without concurrent CTLA-4 inhibition—which enhances T-cell priming and expansion—the immune activation from monotherapy appears inadequate for long-term tumor control. Over time, tumor escape mechanisms, such as the upregulation of other immune checkpoints (e.g., TIM-3, LAG-3), the persistence of regulatory T-cells, or the re-establishment of an immunosuppressive tumor microenvironment, may enable tumor growth. Thus, the initial tumor regression reflects an early immune response to monotherapy, but it was not sustainable without adding further checkpoint inhibitors [7,8]. Similar early, but transient, effects were seen in non-responders, where the tumor volumes decreased shortly after starting therapy, and then resumed growth by days 10–15. PET imaging at day 4 showed increased tumor-to-blood ratios in both the monotherapy group and non-responders, indicating initial immune engagement across these groups. However, unlike responders, this activity did not result in long-term tumor control, likely because of inadequate effector T-cell persistence, the limited expansion of cytotoxic clones, or tumor-specific adaptive resistance mechanisms. Overall, these results suggest that both monotherapy and non-responder tumors exhibit an early immune activation phase, but only some sustain it enough for long-term regression. The ability of ^68^Ga-NOTA-CYT-200 PET to detect these transient responses highlights its potential for monitoring immune activity, even when it does not lead to lasting tumor control. Similar patterns of temporary shrinkage followed by progression have been observed in preclinical and clinical studies of the PD-1 blockade, supporting the plausibility of our findings [7,8,10].

When analyzing subgroups, the treatment responders treated with either combination therapy or monotherapy showed significantly lower tumor volumes 15 days after the treatment (97 ± 47 mm^3^) compared to both the treatment non-responders (311 ± 121 mm^3^) and the controls (301 ± 57 mm^3^, *p* < 0.0001 for both). Importantly, ^68^Ga-NOTA-CYT-200 PET imaging revealed increased TBRs as early as four days after starting the treatment in both therapy groups, with significantly higher TBRs in the responding mice (4.28 ± 1.20) compared to the non-responders (2.73 ± 0.88, *p* = 0.0109) and controls (1.75 ± 0.39, *p* = 0.0002; see Section 2.4 and Figure 4). No significant difference was observed between the non-responders and the controls (*p* = 0.175). Overall, these findings indicate that a higher TBR is strongly correlated with a substantial tumor size reduction, especially in the combination therapy group, where both the highest TBR (4.28 ± 1.20 on days 4–7) and the greatest tumor shrinkage (–64.31 ± 7.71% on day 7) occurred. Interestingly, ^68^Ga-NOTA-CYT-200 PET imaging detected significantly higher TBRs in true responders as early as days 4 or 7 after treatment—before visible tumor shrinkage. This temporal dissociation—where an elevated TBR precedes anatomical changes—highlights the value of GZB PET imaging for early immune monitoring, potentially overcoming the limitations of standard CT or MRI in recognizing pseudo-progression [2]. The ability of ^68^Ga-NOTA-CYT-200 PET imaging to differentiate responders from non-responders early during therapy holds significant clinical implications. While the current biomarkers (e.g., PD-L 1 expression, CD 8 + T-cell density) are limited by spatial and temporal heterogeneity [1,10,11,12], our results demonstrate that granzyme B PET provides a dynamic, functional assessment of T-cell cytotoxicity—as evidenced by higher TBR values predicting subsequent substantial tumor regression. Our data on ^68^Ga-NOTA-CYT-200 PET imaging clearly distinguish immunotherapy responders from non-responders in a humanized mouse model of melanoma, as shown by PET quantification. However, this is a proof-of-concept study; although the promising preclinical results suggest that ^68^Ga-NOTA-CYT-200 PET imaging could effectively identify responders, further research is necessary to validate these findings and develop reliable predictive models before clinical application.

Our study also examined the potential of ^68^Ga-NOTA-CYT-200 PET imaging to detect off-target immune activation associated with IrAEs. We observed an increased ^68^Ga-NOTA-CYT-200 uptake in the liver and intestines of treated mice, especially in the group receiving combination therapy. This aligns with clinical observations, which show that IrAEs commonly affect these organs [14,15,16,17]. Although the LBR varied between groups—likely due to various factors such as the partial hepatobiliary clearance of the tracer—the consistent increase in intestinal tracer accumulation, particularly at day 4 post-treatment, is significant. It suggests that GZB PET imaging could serve not only as a response biomarker, but also as a tool for the early, non-invasive monitoring of IrAEs [18] or inflammatory diseases such as inflammatory bowel disease [4], allowing clinicians to adjust the therapy before symptoms appear. An ex vivo analysis, including biodistribution and immunohistochemistry, confirmed the in vivo PET findings. GZB staining was most prominent in the tumor tissue of the combination group, followed by the monotherapy group, reflecting the in vivo ^68^Ga-NOTA-CYT-200 uptake pattern. The liver and intestinal tissues also showed increased GZB staining in treated animals, supporting the imaging data and highlighting the usefulness of CYT-200 as a whole-body T-cell activity biomarker [18].

We acknowledge some limitations of this study, such as the lack of in vivo kinetic analyses of ^68^Ga-NOTA-CYT-200. Although our imaging protocol used static PET scans at key timepoints (days 4, 7, and 14) to observe immune activation, full kinetic modeling of the tracer was not conducted. This was mainly due to practical constraints in longitudinal small animal imaging and the focus on establishing a proof of concept for treatment response discrimination. In vivo kinetic studies, including dynamic PET imaging with arterial input function measurements or reference tissue modeling, would offer detailed insights into tracer pharmacokinetics, binding specificity, and improve quantitative accuracy. Future studies will aim to include comprehensive kinetic analyses and dose-optimization research to better characterize the biological and pharmacokinetic properties of ^68^Ga-NOTA-CYT-200. These efforts will be crucial for refining imaging protocols and advancing clinical translation. Another limitation is that some off-target binding of ^68^Ga-NOTA-CYT-200 was observed in tumors of the control group, despite its high specificity for active GZB. This was likely due to the ongoing activation of the PBMCs against tumor tissue and mouse organs, which increases over time after PBMC engraftment. Additionally, our results are specific to a melanoma model and need validation in other cancer types. While GZB PET imaging is a common biomarker for immunotherapy responses, it remains uncertain whether ^68^Ga-NOTA-CYT-200 PET imaging is sensitive enough to detect responses to ICI combinations targeting new checkpoints or other immunotherapies such as CAR-Ts and vaccines. Future research should explore dose–response relationships and kinetic modeling, and establish thresholds for clinical decision-making. The clinical application of ^68^Ga-NOTA-CYT-200 will also require initial human testing.

## 4. Materials and Methods

### 4.1. Animal Studies

Fifty-five 8-week-old naïve NSG mice (both males and females) were obtained from Jackson Laboratory (Bar Harbor, ME, USA). The mice were housed in a Biosafety Level 2 facility on a 12 h light–dark cycle with free access to a standard rodent chow diet (approximately 18% protein, 5% fat, and 5% fiber) and water. The body weights at the time of tumor implantation (day 0) averaged 23.62 g ± 4.32. The animals were acclimated for two weeks before the start of the experiments.

### 4.2. Probe Characterization

NOTA-CYT-200 was synthesized (CytoSite Biopharma, Sudbury, MA, USA) with a high chemical purity (>95%) and characterized using mass spectrometry. We performed a fluorometric human GZB activity assay (ab157403) to determine the inhibitor constant (Ki) of NOTA-CYT-200 for active human GZB enzyme activity in vitro. NOTA-CYT-200 was serially diluted with GZB assay buffer, starting from 5000 to 0.00002048 nanomolar. Each dilution was combined with 0.1 µg/µL of active GZB protein (ab285779) and the provided fluorophore. Each well contained a total volume of 50 µL, and all the samples were tested in duplicate using a microplate reader in kinetic mode (BioTek Cytation5, Winooski, VT, USA) after 90 min of incubation at 27 °C. The in vivo specificity of NOTA-CYT-200 labeled with 68gallium (^68^Ga-NOTA-CYT-200) for the active human GZB protein was evaluated in non-tumor-bearing naive NSG mice (n = 8) following a subcutaneous injection of 80 µL of Matrigel alone (Corning™ Matrigel™ Membrane Matrix) or 65 µL of Matrigel containing 15 µL of active human GZB protein (0.05 µg/µL, totaling 0.75 µg GZB) into each animal’s shoulder. Imaging was performed one hour after an i.v. injection of 4.39 MBq ± 0.40 of ^68^Ga-NOTA-CYT-200. The uptake of ^68^Ga-NOTA-CYT-200, measured as standard uptake values (SUVs) within the implanted regions, was assessed using the VivoQuant™ software, version 4.0 (InVicro, Needham, MA, USA). The SUVs were normalized to the blood pool activity by drawing a region of interest around the left ventricle of the heart to calculate the tumor-to-blood ratio (TBR) for further analysis. This method accounts for a nonspecific background signal and enhances the accuracy of the target uptake measurement.

### 4.3. Experimental Design

For immune reconstitution, 2 × 10^6^ peripheral blood mononuclear cells (PBMCs) of human origin were administered intravenously into 10-week-old mice (n = 47), two weeks before tumor implantation. Human melanoma G361 (ATCC^®^) cells were cultured in McCoy’s Medium (Thermo-Fisher, Waltham, MA, USA) supplemented with 10% fetal bovine serum (Thermo-Fisher) and 1% penicillin–streptomycin (Thermo-Fisher) at 37 °C with 5% CO_2_. The mice were then subcutaneously (s.c.) implanted with 2 × 10^6^ G361 cells suspended in Matrigel and 100 µL of phosphate-buffered saline (Sigma-Aldrich, St. Louis, MO, USA) in a 1:1 volume ratio into the left shoulder. The tumor size was measured every 2–3 days using digital calipers with a precision of ±0.1 mm. To estimate the tumor volume, both the longest and shortest perpendicular diameters were measured. These measurements were then combined using a standard method commonly employed in preclinical studies to approximate the tumor volume, providing a reliable estimate of the three-dimensional tumor size. Four days after tumor implantation, the tumors measured 336 ± 38 mm^3^. The mice were then randomly assigned to receive three doses at three-day intervals of either 200 µg/100 µg of an anti-PD1/CTLA4 combination therapy (combination group, n = 15), 200 µg of anti-PD1 as a monotherapy (monotherapy group, n = 15), or a vehicle solution (controls, n = 12) via an intraperitoneal (i.p.) injection. A subset of mice (combination: n = 8, monotherapy: n = 9, controls: n = 6) underwent PET/CT imaging before treatment (baseline), as well as 4 days (timepoint 2), 1 week (timepoint 3), and 2 weeks (timepoint 4) after the treatment began (Figure 7).

### 4.4. Radiolabeling and PET-CT Imaging

^68^Gallium was obtained by eluting a ^68^Ga/^68^Ge generator (Eckert and Ziegler, Berlin, Germany) with 0.1 molar HCl. A total of 2 mL of eluted ^68^gallium was mixed with 100 µL of NOTA-CYT-200, stored in a stock solution of 1 mg/mL of metal-free water (OmniTrace Ultra, Millipore, Burlington, MA, USA) and two molar HEPES buffer to adjust the pH to 3.5–4.0. The reaction was incubated at 95 °C for 10 min. The mixture was then adjusted to a pH of 7.0 using sodium hydroxide (NaOH) before being administered intravenously via the tail vein. A total of 3.96 MBq ± 0.81 of ^68^Ga-NOTA-CYT-200 was injected 60 ± 5 min prior to PET/CT acquisition. The PET/CT images were acquired in static mode using a small animal Argus PET/CT scanner (Sedecal, Madrid, Spain) in two bed positions over 20 min. The PET images were reconstructed using (2 iterations and 16 subsets) 2D-OSEM and corrected for randoms and scatter. Post-acquisition, the PET/CT images were analyzed by two blinded investigators using the VivoQuant™ software, version 4.0 (InVicro). Standard uptake values (SUVs) for the tumor, liver, and colon were normalized to the blood pool activity measured by drawing a region of interest around the left ventricle of the heart, yielding target-to-background ratios (TBR for tumor, LBR for liver, and CBR for colon). This normalization accounts for the nonspecific background signal and enables a comparison of the uptake across the groups and imaging timepoints.

### 4.5. Biodistribution

The ^68^Ga-NOTA-CYT-200 biodistribution was evaluated for each treatment group four days after the initial dose. Major organs, such as the tumor, heart, lungs, liver, spleen, stomach, pancreas, intestines, kidneys, muscles, bones, skin, and brain from each mouse, were excised exactly one hour after the i.v. administration of 52.83 µCi ± 0.87 (controls, n = 3), 41.83 µCi ± 5.93 (mono, n = 4), or 34.65 µCi ± 3.82 (combination, n = 4) ^68^Ga-NOTA-CYT-200. Each organ was weighed individually before measuring the activity with an automatic gamma counter (PerkinElmer Wallac Wizard 2480, Waltham, MA, USA).

### 4.6. Immunohistochemistry

The tissue samples were dissected four days after the treatment to correlate the PET imaging radiotracer uptake with the immunofluorescent staining for GZB. Staining was performed on deparaffinized tissue samples (combination, n = 3; mono, n = 2; and controls, n = 3) using previously described methods [4]. Briefly, GZB staining was conducted with a rabbit anti-human GZB primary antibody (ab243879; Abcam, Cambridge, UK) and Alexa Fluor Plus 647 highly cross-adsorbed goat anti-rabbit IgG secondary antibody (a32733; Thermo Fisher). Microscopy images were obtained by the Brigham Women’s Hospital Pathology Core Services, and quantification was performed using the ImageJ2 (version 2.9.0; Fiji) software.

### 4.7. Statistical Analyses

Prism (version 9.2; GraphPad) was used to perform quantitative data analyses. The data are presented as the mean ± SEM. The means of the two groups were compared using unpaired Student’s t-tests when appropriate. One- or two-way ANOVA with Tukey’s multiple comparison test was used as needed to compare the differences among the treatment groups. *p*-values of <0.05 were considered statistically significant.

## 5. Conclusions

In conclusion, our study demonstrates that ^68^Ga-NOTA-CYT-200 PET imaging enables the early, specific, and quantitative detection of immune-mediated tumor cytotoxicity in a humanized melanoma model. It offers significant advantages over current methods by providing a spatially resolved, non-invasive assessment of both the treatment efficacy and the IrAEs. These findings support the further development and clinical validation of CYT-200 as a dual-purpose imaging biomarker for immunotherapy monitoring. If validated in patients, this approach could facilitate more precise, timely, and individualized immunotherapy management.

## Figures and Tables

**Figure 1 pharmaceuticals-18-01309-f001:**
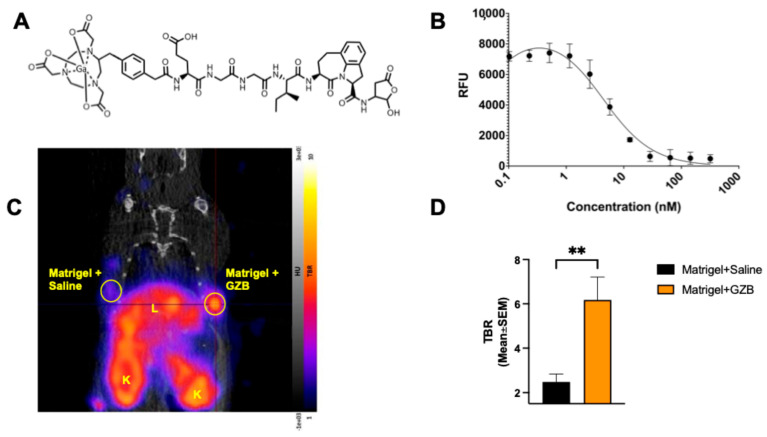
(**A**) Chemical structure of the tracer, ^68^Ga-NOTA-CYT-200. (**B**) Enzyme inhibition graph demonstrating the inhibition of the enzymatic activity of the recombinant human GZB protein by NOTA-CYT-200 in a dose-dependent manner. (**C**) Representative ^68^Ga-NOTA-CYT-200 PET/CT image of a mouse with an implanted subcutaneous Matrigel + GZB implant in the left upper flank and a Matrigel + saline implant in the right flank, clearly showing a much higher uptake in the left flank compared to the other side. L = Liver. K = Kidney. (**D**) There was a significantly greater (** *p* < 0.01) uptake of ^68^Ga-NOTA-CYT-200 in the Matrigel + GZB implants compared with the implants containing Matrigel + saline, confirming the target specificity of ^68^Ga-NOTA-CYT-200 in vivo.

**Figure 2 pharmaceuticals-18-01309-f002:**
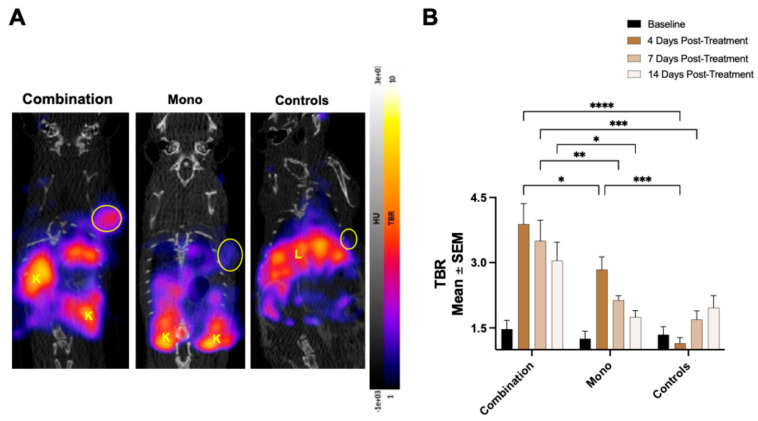
(**A**) Representative ^68^Ga-NOTA-CYT-200 PET/CT images on day 4 after treatment administration, showing a visibly increased uptake in the combination treatment group, followed by the monotherapy and control groups. (**B**) The mean ^68^Ga-NOTA-CYT-200 PET TBR was similar in all groups at baseline and was significantly increased 4 days after treatment initiation in both groups receiving either combination therapy or monotherapy. The TBR was greatest in the combination group on days 4, 7, and 14 after treatment initiation. ^68^Ga-NOTA-CYT-200 TBR was also significantly higher on days 7 and 14 post-treatment initiation in the combination group. * *p* < 0.05, ** *p* < 0.01, *** *p* < 0.001, **** *p* < 0.0001.

**Figure 3 pharmaceuticals-18-01309-f003:**
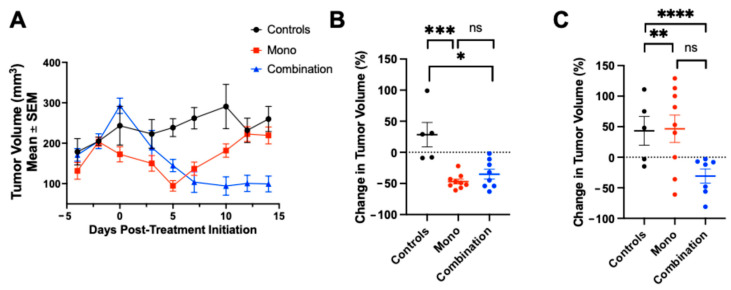
(**A**) Tumor growth curves for all treatment groups over 15 days following treatment initiation. (**B**) On day 5, both the monotherapy and combination groups exhibited significantly lower tumor volumes compared to the controls (control vs. mono, *p* = 0.0002, ***; control vs. combination, *p* = 0.0255, *). No significant difference was found between the monotherapy and combination groups. (**C**) On day 7, both treatment groups again showed significantly reduced tumor volumes compared to the controls (control vs. mono, *p* = 0.0013, **; control vs. combination, *p* < 0.0001, ****), with no significant difference between the monotherapy and combination groups. By day 15, the combination group maintained a notable reduction in tumor volume compared to the controls, while tumors in the monotherapy group regrew, resulting in no significant difference between the monotherapy group and the controls at the final timepoint.

**Figure 4 pharmaceuticals-18-01309-f004:**
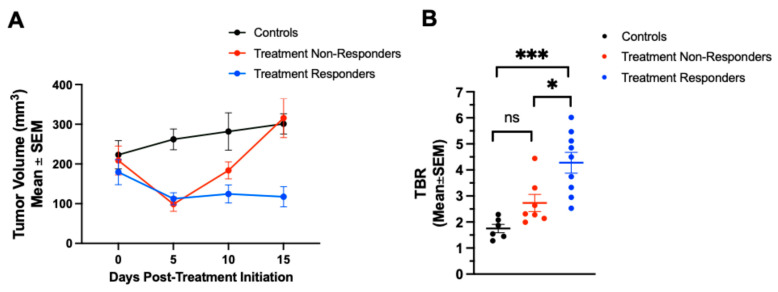
(**A**) Tumor growth curve (tumor volumes in mm^3^) showing the separation of TRs and TNRs from the controls, regardless of the treatment type (combination or monotherapy), up to 15 days after starting treatment. (**B**) TBRs were measured through PET imaging on day 4 or 7 after treatment began. TRs exhibited significantly higher TBR values compared to the controls and TNRs. No difference was observed between the controls and TNRs. ns: not significant; * *p* ≤ 0.05, *** *p* < 0.001; two-way ANOVA with Tukey’s multiple comparisons test.

**Figure 5 pharmaceuticals-18-01309-f005:**
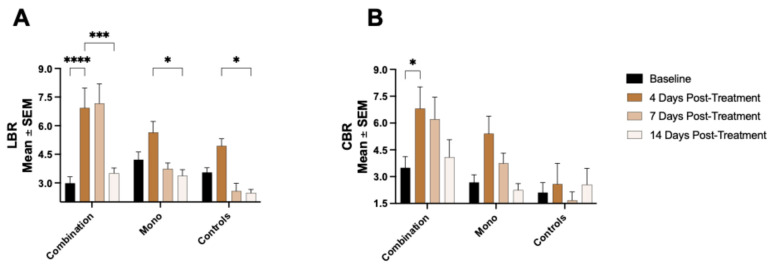
(**A**) The intestinal ^68^Ga-NOTA-CYT-200 uptake was highest 4 days after starting treatment in the combination group, followed by the monotherapy group; however, the difference was only statistically significant for the combination group. Both groups showed a gradual decrease in the CBR on days 7 and 14. The CBRs in the control group remained consistent across all imaging timepoints, both before and after treatment initiation. (**B**) Similarly, the LBR was significantly higher at 4 and 7 days after treatment compared to baseline; however, there was no significant difference in the LBR between any post-treatment imaging timepoints and baseline. ns = not significant; *p* > 0.05; * *p* ≤ 0.05; *** *p* ≤ 0.001; **** *p* ≤ 0.0001; two-way ANOVA with Tukey’s multiple comparisons test.

**Figure 6 pharmaceuticals-18-01309-f006:**
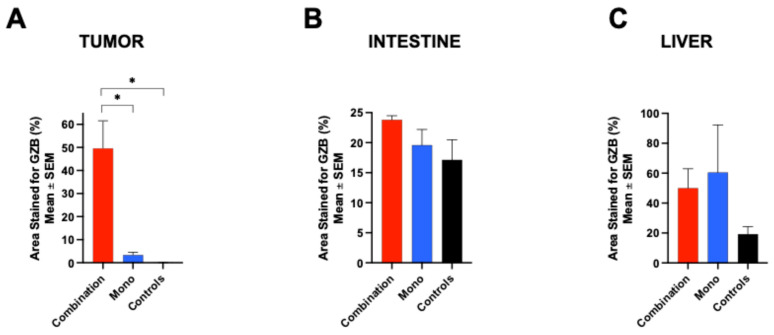
Immunohistochemical staining was conducted on tumor (**A**), intestinal (**B**), and liver tissue (**C**) samples collected 4 days after tumor implantation. The most intense GZB staining in the tumor and colon tissues appeared in the group receiving combination therapy, followed by the monotherapy group. GZB staining was significantly greater in the combination group compared with the monotherapy and the control group, both of which presented with similar GZB staining. The control group exhibited minimal or no staining in the tumor tissues (**A**,**B**). Similarly, the GZB staining levels in the liver samples were higher in the combination and monotherapy groups compared to the controls. However, there was no significant difference in GZB staining of intestinal and liver tissue between the treatment groups (* *p* ≤ 0.05; two-way ANOVA with Tukey’s multiple comparisons test) (**B**,**C**).

**Figure 7 pharmaceuticals-18-01309-f007:**
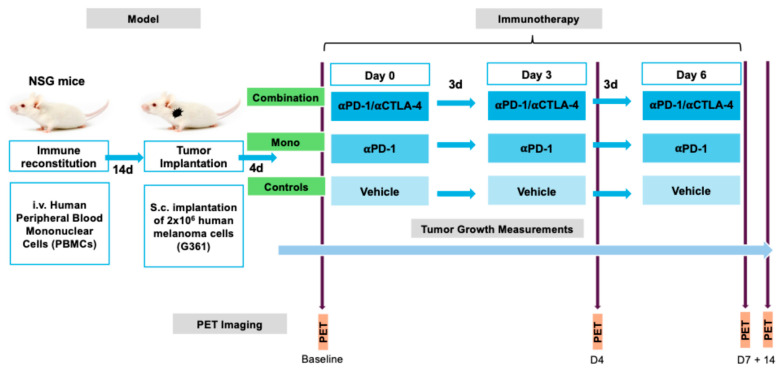
Experimental design. Naïve NSG mice received a tumor implantation 14 days after receiving human-derived PBMCs. The mice were then randomly assigned to one of three treatment groups: a combination of checkpoint inhibitors (combination), treatment with a single checkpoint inhibitor (mono), or a vehicle solution (controls). Tumor growth was monitored regularly until the final timepoint. All groups were imaged using PET-CT before the treatment began (baseline) and on days 4 (D4), 7 (D7), and 14 (D14) after the initial dose.

## Data Availability

Data presented in this study is contained within the article.

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
