# Peer review of "Granzyme B PET Imaging Enables Early Assessment of Immunotherapy Response in a Humanized Melanoma Mouse Model"

_pharmaceuticals, 2025, doi:10.3390/ph18091309_

Round 1

Reviewer 1 Report

Comments and Suggestions for Authors

The authors present a compelling preclinical study with an ​extensive body of work. While the manuscript has significant merit and appears suitable for publication upon addressing the following concerns, I have several points requiring clarification before final acceptance can be recommended.

​Major Comments:​​

​1. Discrepancy in PET Uptake:​​ The authors state that treatment resulted in increased uptake of ⁶⁸Ga-NOTA-CYT-200 PET in the liver and intestine. However, Figure 2A appears to contradict this, showing noticeably ​lower liver uptake​ in the mice receiving combination therapy and monotherapy compared to the control group in the representative images. This apparent discrepancy requires resolution. Please explain this inconsistency clearly.

​2. Inconsistency in Tumor Volume Data:​​ In section 2.3, line 122, the authors state there was ​​"no significant change"​​ in tumor volume among the three groups within the first 7 days post-treatment. However, the tumor growth curves depicted in Figure 3A ​seem to visually contradict this claim, suggesting differences may be present even if deemed non-significant. Please reconcile this statement with the data presented in Figure 3A or provide supporting quantitative analysis (e.g., statistical comparison specifically for day 7).

​3. The authors should provide a more ​in-depth discussion​ of two intriguing observations in the tumor volume data:

​3.1 Mono therapy groups:​​ Figure 3A shows that the ​mono subgroup exhibited a transient reduction in tumor volume within the first 5 days​ post-treatment, followed by subsequent regrowth. The underlying mechanism for this initial response and subsequent escape deserves elaboration.

​3.2 TNR Group:​​ Similarly, in Figure 4A, the ​TNR subgroup also shows an initial reduction in tumor volume within the first 5 days. The reason(s) for this early response in the TNR subgroup warrants discussion alongside the monotherapy observation.

Minor Comments:​​

​1. In panels C and D of Figure 1, the Matrigel injection control group should be clearly labeled as ​​"Matrigel + Saline"​​ in the figure legend, rather than simply "Matrigel".

​2. Clarification of Group Sizes:​​ As the Methods section follows the Results, it would greatly aid reader comprehension if the authors explicitly state the ​number of animals per group (n=X)​​ each time the combination therapy group, monotherapy group, and control group are mentioned within the Results section.

​3. Please provide the ​scientific rationale​ for choosing Days 4, 7, and 14 as the specific time points for PET/CT imaging evaluations.

​4. Figure 2A should include a ​colorbar​ to allow interpretation of signal intensity in the PET images, as was done in Figure 1C.

​5. An ​extra apostrophe (')​​ is present on line 335. Please remove it.

Reviewer 2 Report

Comments and Suggestions for Authors

Checkpoint inhibitor immunotherapy is a key tool in cancer treatment, and humanized mice, harboring both the human immune system and human tumors, represent a valuable preclinical model for cancer immunotherapy research, allowing for the testing of novel immunotherapies and their combinations. The translational value of highly immunosuppressed mice engrafted with human immune cells and either a cultured human cell line or patient-derived cells has been recognized, and efforts to further improve their representativeness are underway. 

In several types of tumors, only a limited number of patients respond to immune checkpoint inhibitors, then  monitoring early responses to these inhibitors is an urgent need. Over the past decade, researchers, including some of the manuscript authors, have investigated the use of new PET tracers based on 68 Ga-NOTA and targeting Grazyme B, a downstream effector of tumoral cytotoxic T cells, to detect immune system activation in mouse models of  inflammatory bowel disease (IBD), colon cancer and breast cancer, as well as early response to combined immunotherapy in patients with gastric cancer and on human melanoma specimens.

In this manuscript, the authors provide a preclinical proof-of-concept aimed at evaluating the utility of a novel 68Ga PET probe targeting granzyme B as a biomarker for early prediction of tumor response to immunotherapy and immune-related adverse events in a humanized mouse model of melanoma.

Background/Objectives are clearly stated in the abstract, while the other sections are short (I think due to the word count requirement) but comprehensive.

Regarding the text, I would like to invite the authors to carefully review the manuscript and provide some suggestions to further improve its quality:

line 34-35: "Immunotherapies have significantly impacted cancer therapy over the past decade and have fundamentally changed how cancer is treated [1-4]." The authors repeat the same concept in the same sentence, please rewrite it.

line 34-35: please, check this suggestion:

"While the estimated proportion of cancer patients eligible for immune checkpoint inhibitor (ICI) therapy has increased by a factor of 30 over nine-years, rising from 1.54% in 2011 (95% confidence interval (CI) 1.51–1.57%) to 43.63% in 2018 (95% CI, 43.51–43.75%), )  there are some major limitations associated with ICI therapies [5].

line 45-47: the "term "tool" is repeated in close proximity, in two sentences expressing mainly the same concept, please rewrite it.

line 48-61: I would suggest that the authors discuss more extensively the advantages of imaging over biopsy (invasiveness, sensitivity, 3D, etc.), especially molecular imaging, as well as the relative limitations that drive the development of new radiotracers for this specific problem. Furthermore, I would suggest to explain briefly the main characteristics of the NOTA-CYT-200 molecule before aims statement, explaining the reasons/utility for its choice/testing and the advantages compared to other enzyme granzyme B targeting probes.

line 46-50: the "term "To date"  is repeated closely; please find a synonym or rewrite a sentence to improve the clarity of the manuscript.

line 68-73, section 2.1.: Could the authors discuss the potential implications of NOTA-CYT-200 inhibiting the active enzyme activity of GZB in section 3 "Discussion"?

line 72-76: lease check the clarity of the sentences or any typos; the explanations referring to the letters in Figure 1 do not correspond to the figures themselves and the captions in Figure 1. I would also suggest to write in the text  "Matrigel  implant", not Matrigel-tumors, as in discussion, line 266.

line 94-97: "The Combination (group?)"...Monotherapy.....the Control....(ones?). The same suggestions for section 2.3. To simplify the consistency of the text, I would suggest using an abbreviation for each group, indicated the first time they are mentioned.

line 131-132: I would suggest to rewrite the sentence to improve clarity, e.g "While the combination group showed a progressive reduction in tumor size over time up to 15 days after treatment induction,..."

line 144-145: "...mice that exhibited tumor size reduction at only one (earlier?) of the two timepoints...

line 167-168: Figure 4 is displaced.

line 214-216: please clarify the concept: "four days after treatment initiation, which represented the time point of highest tracer expression measured in vivo"

line 219-223: are the authors referring to the kidneys?

line 232-241: please, improve clarity of the sentences: the authors performed immunofluorescent GZB staining on tumor, intestine, and liver tissue samples from all three groups, which were collected on day 4.  In lines 232-236 it is not clear which organ the authors are referring to (tumor?).

Overall, in section 2.5. Treated mice show greater GZB expression in the liver and intestine and  2.6. Ex vivo measurements of GZB correlate with in vivo findings, I would suggest simplifying the text and emphasizing the usefulness/informativeness of figures/captions, integrating a brief explanation of how the reported results might biologically implicate.

line 252-253: I would suggest to rewrite the sentence to improve clarity, e.g. "providing a living experimental system featuring a functional human tumor-immune microenvironment, with increased translational relevance".

line 266-268: I would suggest clearly specifying that this is a proof-of-concept study, and while the encouraging preclinical results suggest the utility of this radiotracer in distinguishing between responders and non-responders, further studies are needed to develop an effective predictive model from comparable data.

line 350: 4. Materials and Methods section 4.1. Animal Studies, line 386-388, 4.3. Experimental Design:  Please, improve animal signalment (diet composition, gender, age and body weight at the time of the experiment).

line 364-372: please, provide technical details for PET acquisition (PET scanner, resolution, sensitivity, FWHM, static or dynamic acquisition, frame sequence duration, specific radioactivity of radiotracer, reconstruction algorithm, voxel size, random, scatter, dead time, and decay correction, etc). Furthermore, I would suggest rewrite sentence line 171-172, e.g: SUV values were normalized  to the blood pool of the heart's left ventricle as the target-to-background ratio (TBR): usually, a region of interest is  drawn around the left ventricle of the heart to calculate blood pool activity as a measure of nonspecific signal. The same suggestion would be for line 405-407.

line 381: Please add details on the accuracy of the caliper measurements and the method used to calculate the volume.

line 373-394, section 4.3. Experimental Design: please, provide details about the age of mice at each experimental procedure (at least PBMCs and C361 cells injection).

line 15, 64, 96, 216, 289, 418: please, rewrite where "radiotracer expression" is reported. Radiotracers are taken up by tissues and organs, with increased activity/concentration/tracer uptake where they bind to specific molecules or track specific molecular processes, allowing their expression/activity to be visualized. The  term "expression" in line 175, 236, 256, 317 properly refers to the biomarkers expression.

 Could the authors explain why in vivo kinetic studies have not been performed (also as limitation)?

Comments on the Quality of English Language

The English is fine, but in several parts of the text the clarity of the results could be improved.
